# Generation of Population-Level Diversity in *Anaplasma phagocytophilum msp2/p44* Gene Repertoires Through Recombination

**DOI:** 10.3390/pathogens14030233

**Published:** 2025-02-27

**Authors:** Anthony F. Barbet, David R. Allred, Francy L. Crosby

**Affiliations:** 1Department of Infectious Diseases and Immunology, University of Florida, Gainesville, FL 32611-0880, USAfc675@missouri.edu (F.L.C.); 2Emerging Pathogens Institute, University of Florida, Gainesville, FL 32611-0880, USA; 3Genetics Institute, University of Florida, Gainesville, FL 32611-0880, USA

**Keywords:** tick-borne diseases, antigenic variation, repertoire variability, anaplasmosis, emerging diseases, *msp2/p44*

## Abstract

*Anaplasma phagocytophilum*, a tick-borne Rickettsiales, causes an emerging disease among humans and animals called granulocytic anaplasmosis. The organism expresses an immunodominant surface protein, MSP2/P44, that undergoes rapid antigenic variation during single infections due to gene conversion at a single genomic expression site with sequences from one of ~100 transcriptionally silent genes known as “functional pseudogenes”. Most studies have indicated that the predominant gene conversion mechanism is the insertion of complete central variable regions (CVRs) into the *msp2/p44* expression site via homologous recombination through 5′ and 3′ conserved regions. This suggests that it is possible that persistent infections by one strain may be self-limiting due to the exhaustion of the antigenic repertoire. However, if there is substantial recombination within the functional pseudogene repertoires themselves, it is likely that these repertoires have a high rate of change. This was investigated here by analyzing the repertoires of *msp2/p44* functional pseudogenes in genome-sequenced *A. phagocytophilum* from widely different geographic locations in the USA and Europe. The data strongly support the probability of recombination events having occurred within and between *msp2/p44* repertoires that is not limited to the 5′ and 3′ conserved regions of the CVR, greatly expanding the total potential variation. Continual variation of *msp2/p44* repertoires is predicted to aid the organism in overcoming existing immunity in the individual and causing superinfections among immune populations, and this may facilitate the adaptation of the microorganism to infect and cause disease in different species.

## 1. Introduction

*Anaplasma phagocytophilum* is a Gram-negative, tick-borne, obligate intracellular bacterium that is transmitted by hard-bodied ticks of the *Ixodes* species and is found worldwide, particularly in the northern hemisphere, as it follows the distribution of its vector [1]. In the US, Europe, and Asia, this pathogen causes an emerging and potentially fatal disease in humans known as human granulocytic anaplasmosis (HGA). In the US, case reports of this disease have increased steadily, from 273 cases in 2000 to 5651 in 2022 (https://www.cdc.gov/anaplasmosis/hcp/statistics/index.html accessed on 20 February 2025). HGA may range from mild, flu-like symptoms of fever, headache, and myalgia to death [2,3], with a reported hospitalization rate of 36% [4] and mortality rates varying from about 1% in the USA to 26.5% in China [5,6]. These wide differences in mortality indicate potentially significant differences among the predominating strains of the organism. *Anaplasma phagocytophilum* is also a pathogen of high importance in veterinary medicine, as it can infect a wide variety of hosts, ranging from wild to domestic animals (rodents, ruminants, dogs, horses, etc.) [1], causing a form of granulocytic anaplasmosis that is referred to by a variety of names depending upon the infected host. In Norway, this agent has long been a barrier to domestic sheep production by causing abortion, diminished fertility, and reduced weight gains [7]. *A. phagocytophilum* is transmitted biologically by Ixodid ticks, whose metabolism and innate immunity it modifies to facilitate successful infection [8,9]. The tick vector’s own feeding preferences likely affect the distribution of the bacterium and the probability of exposure of a given mammalian host to the organism.

Despite the organism having a reduced genome, *A. phagocytophilum* has increased genetic diversity, characterized by the presence of several strains or variants displaying different host predilections, and not all strains infect all hosts [10,11]. Animals that recover from acute disease will become persistent carriers of this bacterium. This characteristic is thought to be enabled in part by the sequential expression of variable surface antigens encoded by the *msp2/p44* multigene family [12]. As in many systems of antigenic variation, the antibody to *msp2/p44* neutralizes the infection caused by the homologous serotype. However, structurally different MSP2/P44 proteins are created and dominate in different organism peaks during infection; these are only recognized by antibodies generated subsequent to their appearance [13,14,15,16]. The demonstrated basis for antigenic variation is the insertion of a central variable region (CVR) into an *msp2/p44* genomic expression site by gene conversion, utilizing the RecF recombination pathway [17,18,19,20]. Different CVRs are present in numerous copies in the *A. phagocytophilum* genome, flanked by conserved 5′ and 3′ sequences. Genome sequencing of the HZ strain identified 113 copies of *msp2/p44*, some of which did not contain either or both of the 5′ and 3′ conserved regions [21]. In *Anaplasma marginale*, which has a similar system for the variation of *msp2*, there are only seven or eight copies of the *msp2* CVR (termed “functional pseudogenes”) available for insertion into the single expression site [22,23]. In that species, additional variation is achieved by the use of different short regions from the CVRs to form complex mosaics, particularly in long-term persistent infections [24,25]. In serial infections with *A. phagocytophilum* in a mouse model, among 263 expressed pseudogenes, only three mosaics were detected and these involved contributions from only two different pseudogenes in each case [26]. This agrees with previous data suggesting that the primary mechanism of gene conversion is the insertion of a complete CVR into the expression site [27]. This was confirmed using cloned *A. phagocytophilum* to infect horses and SCID mice that showed recombination break points only in the conserved 5′ and 3′ regions of *msp2/p44* copies, suggesting that the *msp2/p44* antigenic repertoire is limited [18]. This has led to the idea that long-term infections with *A. phagocytophilum* could be self-limiting, due to the exhaustion of the *msp2/p44* repertoire, unless there is an accompanying variation in the repertoire itself [26,27]. The availability of 28 genome-sequenced strains of *A. phagocytophilum* from different geographic locations [11] presents an opportunity to examine this possibility at the population level. Specifically, is there evidence for recombination among members of the *msp2/p44* repertoire leading to the generation of diversity in the repertoires themselves? We provide here evidence of such recombination that may help to explain the superinfections and persistence of this microorganism and perhaps its adaptation to novel hosts.

## 2. Materials and Methods

### 2.1. Selection of A. phagocytophilum Strains for Analysis

To assess for recombination, we employed the strategy of analyzing for evidence of recombination among the silent *msp2/p44* repertoires of six highly divergent strains of *A. phagocytophilum*. We chose to use sequences derived from *A. phagocytophilum* strains infecting different geographically separated mammalian hosts. Our premise was that strains from such divergent, isolated sources would be unlikely to demonstrate evidence of inter-strain recombination due to the isolation of the host and tick populations, but would still allow the detection of intrastrain recombination if it occurred. This also allowed us to ask whether inter-strain recombination, if detected, occurred only between strains infecting the same mammalian host. Previously, the individual *msp2/p44* genes comprising the repertoires of 28 genome-sequenced strains of *A. phagocytophilum* were characterized (provided in Appendix A of Ref. [11]). Those genes sharing at least 99% identity at the nucleotide level were identified and discarded from further analysis, enabling comparisons of the overall variability in the non-identical members of the repertoires. Some strains had nearly identical repertoires, whereas other strain repertoires were completely different, and these differences were based partly on the geographic origin of each strain. For example, two strains isolated from humans in New York state shared most of their repertoires, whereas these two strains shared only ~50% of their repertoires with human-derived strains from the Midwest USA, and a horse-derived strain from Minnesota shared only ~1% of its repertoire with one from California. In the present study, the *msp2/p44* repertoires (Appendix A) of two diverse human strains from New York (HZ2_NY) and Wisconsin (ApWebster_WI), two horse strains from Minnesota (Horse1_MN) and California (Horse1_CA), and two sheep strains from distinct regions of Norway (NorShV1 and ApSheep_NorV2) were selected for analysis.

### 2.2. Determination of msp2/p44 Repertoires

The repertoires of the *msp2/p44* genes in each genome-sequenced strain were determined as described [11]. Briefly, we used an 11-nucleotide sequence present in the 5′ conserved sequence of *msp2/p44* to extract all instances of this sequence plus the downstream 469 nucleotides from all *A. phagocytophilum* genomes. A filter was then applied to verify that each gene encoded at least one of the following known protein characteristics: N-terminal KELAY and N- or C-terminal LAKT amino acid motifs. The 113 msp2/p44 gene loci previously described in the HZ strain [21] include genes characterized as either full-length, silent/reserved, truncated, or fragments. The above methods detected 83 *msp2/p44* genes in our re-sequenced HZ2 strain (accession #CP006616; designated HZ2_NY herein) and could not detect partial genes with no 5′ or 3′ conserved region, thought to be necessary for recombination into the MSP2 expression site. These selection criteria were similarly applied to *msp2/p44* genes from the human-derived Web_WI strain (accession #LANS00000000; designated ApWebster_WI) (a total of 166 genes); two horse-derived derived strains, Horse1_CA (accession #FLMF00000000) and Horse1_MN (accession #FLMC00000000) (166 genes); and two Norwegian sheep-derived strains, NorShV1 (accession #CP046639) and ApSheep_Norv2 (accession #CP015376) (172 genes). In total, 504 *msp2/p44* gene sequences were available for analysis. All sequences are provided in Appendix A.

### 2.3. Detection of Recombination

Sequences were aligned with the CLC Bio proprietary multiple sequence alignment module (break cost = 10, cost to extend = 1) for ease of alignment editing. Alignments were manually optimized prior to use in the analysis of recombination. Recombination detection was performed on the alignments using RDP5 v. 5.64 software [28]. For consistency with the demonstrated mechanism of gene conversion [17,18,19,20], the GENECONV module implemented within RDP5 was employed for the analytical screening of all samples, providing multiple comparisons of the linear sequences with Bonferroni correction and a highest acceptable *p*-value of 0.05. Individual samples were further analyzed with the integrated modules RDP [29], Bootscan [30], Maxchi [31], Chimaera [32], SiSscan [33], PhylPro [34], LARD [35], and 3Seq [36] for the identification of recombination events and/or breakpoint sites, as implemented within RDP5. Modules were adjusted for sensitivity at observed nucleotide change rates. The output of all detected recombinants and the statistical support for them is provided in an Excel-compatible file in Appendix A. The breakpoint density plots utilized a sequence window size of 100 nucleotides and 1000 permutations to infer the existence of statistically supported recombination hot- and coldspots. These are presented herein as breakpoint p-density plots of probabilities, in which 99% (dark grey) and 95% (light grey) confidence intervals are also shown as shaded areas. Hotspots for recombination are inferred where the black plot lines emerge above the shaded areas, and corresponding areas of low recombination (coldspots) are suggested by plot lines dropping below the shaded areas.

### 2.4. Polypeptide Structural Comparisons

The *msp2/p44* repertoire sequences were translated into predicted MSP2 polypeptides in reading frame 1. Sequences were submitted to the Robetta server (http://robetta.bakerlab.org/ accessed on 25 January 2025) for structural prediction, using the Robetta algorithm [37]. Structural predictions were saved as .pdb files and visualized with the UCSF Chimera software v. 1.14 [38]. Superimpositions of the predicted structures were accomplished with the Matchmaker module of Chimera.

## 3. Results

The alignment of the gene repertoires from human-, horse-, or sheep-derived strains of *A. phagocytophilum* identified similar conserved and variable regions in each case (Appendix A), suggesting that the overall structures of the genes comprising these repertoires were consistent and maintained across strains. The conserved and variable regions also conformed to what has been observed previously in different expressed *msp2/p44* cDNAs found in human patients experiencing infection with *A. phagocytophilum* [20]. Indeed, in the alignments of the *msp2/p44* genes of all strains, the conservation of the structure and flanking sequences is clear (Appendix A). The longest conserved regions were in the 5′ and 3′ flanking regions of genes that have been identified previously as the preferred sites for recombination into the *msp2/p44* expression site as part of antigenic variation [18]. Comparing the repertoires of the two human-derived strains from either New York state or Wisconsin, which shared 54% of their repertoires, confirmed the 5′ flanking region as a hotspot for recombination (Figure 1A). Although the 3′ flanking region was not an obvious recombination hotspot in this analysis, examples of recombination were observed there (e.g., Figure 1A, HZ2_NY2014 alignment). Moreover, in an analysis of all *msp2/p44* genes included in this study (Appendix A), very strong statistical support for the 3′ recombination hotspot was obtained (Figure 2). The recombination detected was both between individual genes present in the same repertoire and between genes present in either the New York or Wisconsin strains. Interestingly, the same gene in the HZ2_NY repertoire (1399) appeared to have contributed segments to at least two different copies (2026 and 2059) in the Web_WI repertoire. Putative recombinants also extended beyond the 3′ conserved flanking region into the 3′ variable region (e.g., recombinant HZ2_NY1391; Appendix A). A similar result was obtained when comparing the *msp2/p44* gene repertoires found in the two horse-derived strains of *A. phagocytophilum* from either Minnesota or California, although, in this case, the clear presence of a 3′ recombination hotspot was strongly supported statistically (Figure 1B). In the two sheep-derived strains from different regions of Norway, the putative recombination events appeared to be more complex and extended further into 3′ variable regions of the repertoires. Similar to the human isolates, the 3′ recombination hotspot was not obvious, perhaps because of the resolution of recombination intermediates over a longer region (Figure 1C). In all strains, the sequences showed evidence of prior recombination with unknown *msp2/p44* gene forms that were not recovered among the specific genomes sequenced, providing evidence of additional undefined diversity among the *A. phagocytophilum* strains circulating in the environment. Significantly, it was possible to detect high-probability recombination events between all human, horse, and sheep strains, in all combinations (Appendix A). Interestingly, high-probability recombination events were detected between Norwegian sheep strain genes and those of the HZ2_NY strain and, in this case, involved the 3′ conserved sequences (Figure 3). The finding of recombination events among geographically broadly distributed strains indicates that there is conservation of variable sequence elements in the silent repertoire during the geographic distribution of this agent, as well as their recombination to broaden diversity. In all scenarios, the 5′ conserved flanking region was observed to be a hotspot for recombination, and a coldspot with a low probability for recombination was maintained immediately 3′ to the 5′ hotspot. The relative inconsistency of the 3′ conserved flanking region as an obvious recombination hotspot is curious and seems to be associated with the host species from which the *A. phagocytophilum* strains were isolated. This may reflect the greater or lesser importance of sequences in this region of the MSP2 protein for interactions with specific hosts, resulting in differences in the levels of immune selection and the retention of recombinants altered in this region.

## 4. Discussion

This study demonstrates the outcomes of recombination events occurring between *msp2/p44* CVR gene regions that are largely isolate-specific. The circumstances under which these events occurred are unknown. Moreover, it is important to realize that it is not possible from these studies to identify sequences as being parental or recombinant in origin with certainty, as the evolutionary histories of these strains are unknown. From prior genomic analyses [11], however, it is clear that many USA strains infecting humans, dogs, and horses from the Northeast and Midwest are closely related. In these closely related strains, the recombination analysis suggests that the initial recombination events are into the 5′ and 3′ conserved regions (hotspots) flanking the CVR. In the more distantly related strains, the recombination events are more complex and can extend into the 3′ variable region. The reasons for this polarity are not clear but may be related to gene orientations relative to, and the distances from, the origin of replication. In a prior repertoire analysis [39] that required >90% amino acid identity, rather than >99% nucleotide identity as in the current study—a much lower threshold—more repertoire genes were found to be shared. This suggests that point mutations as well as recombination in *msp2/p44* cause stepwise evolutionary changes that can lead progressively to entirely different surface antigen repertoires. It is not apparent from these analyses whether the mechanism of recombination among *msp2/p44* genes proceeds directly between unexpressed genes in the repertoire, via a multiple step mechanism involving genes present in the expression site specifically, or some measure of both. Gene conversion during antigenic variation normally involves the replacement of transcribed sequences in the expression site with duplicated sequences from the silent repertoire. However, at a much lower frequency, it is likely that this event, which is a form of DNA repair, may proceed in the reverse direction, resulting in the insertion of novel sequence combinations into the silent repertoire.

There are several potential practical implications of the above analyses. First, unlike the gene conversion of the *msp2/p44* expression site by different CVRs occurring during a single infection, such repertoire changes are expected to be more permanent and may facilitate the adaptation of the organism to different tick- and animal host species. For example, the structures of the polypeptide sequences encoded by gene copies ApHZ2_NY1445 (minor parent) and ApWebster_WI2017 (recombinant; Figure 1A) are nearly identical, as predicted by Robetta (http://robetta.bakerlab.org/ accessed on 25 January 2025, [37]) (Figure 4A). By contrast, the major parent, ApWebster_WI2064, differs in structure significantly (Figure 4B), with β-sheet and α-helix structure in what is a region of only random coil in the other two polypeptides. Recombination may result in novel structural combinations that could not only affect immune processing and recognition but also the capacity of the protein to interact with alternative host components if incorporated into the expression site and expressed as MSP2 protein. As MSP2 is thought to have an adhesin function [40], this may help to explain why *A. phagocytophilum* is currently an emerging disease in humans yet has been known to infect sheep for >200 years [7]. Second, there are epidemiologic implications regarding how an endemically stable situation, where most animals become persistently infected, may be rendered unstable by the introduction of *A. phagocytophilum* strains with different repertoires. Observations consistent with this possibility have been made for infections of cattle caused by *A. marginale* [20,41,42]. Third, the identification of less favored intragenic sites for recombination may imply a requirement for the conservation of structure in these regions of the MSP2/P44 proteins, although, given the diversity of the sequences found there and the structural effects of these differences, this would seem unlikely. Selection and retention for the poor immunogenicity of the encoded polypeptide, and the susceptibility to and repair of double-stranded DNA breaks during the replication of these sequences, are alternative possibilities that may also play roles. Fourth, orthologs of *msp2/p44* exist in other species taxonomically related to *Anaplasma* (the Pfam01617 family). For example, *Ehrlichia* spp. have 17–22 tandemly arranged members of this family [21] that are differentially expressed in situ, rather than by recombination into a separate expression site [43,44,45]. These genes express immunodominant surface antigens that have been considered as targets for vaccination [46,47,48]. It is possible that recombination among these gene copies may also occur and influence protective immune responses against heterologous strains. While it is not clear why this gene family is susceptible to DNA damage and rearrangements in some species but not others, this difference has ramifications for their suitability in immunization strategies. The work presented herein demonstrates the susceptibility of *A. phagocytophilum msp2/p44* to recombination within the silent *msp2/p44* repertoire outside of antigenic variation, and it may help to explain both microbial persistence at the host population level and adaptation for the infection of novel hosts.

## Figures and Tables

**Figure 1 pathogens-14-00233-f001:**
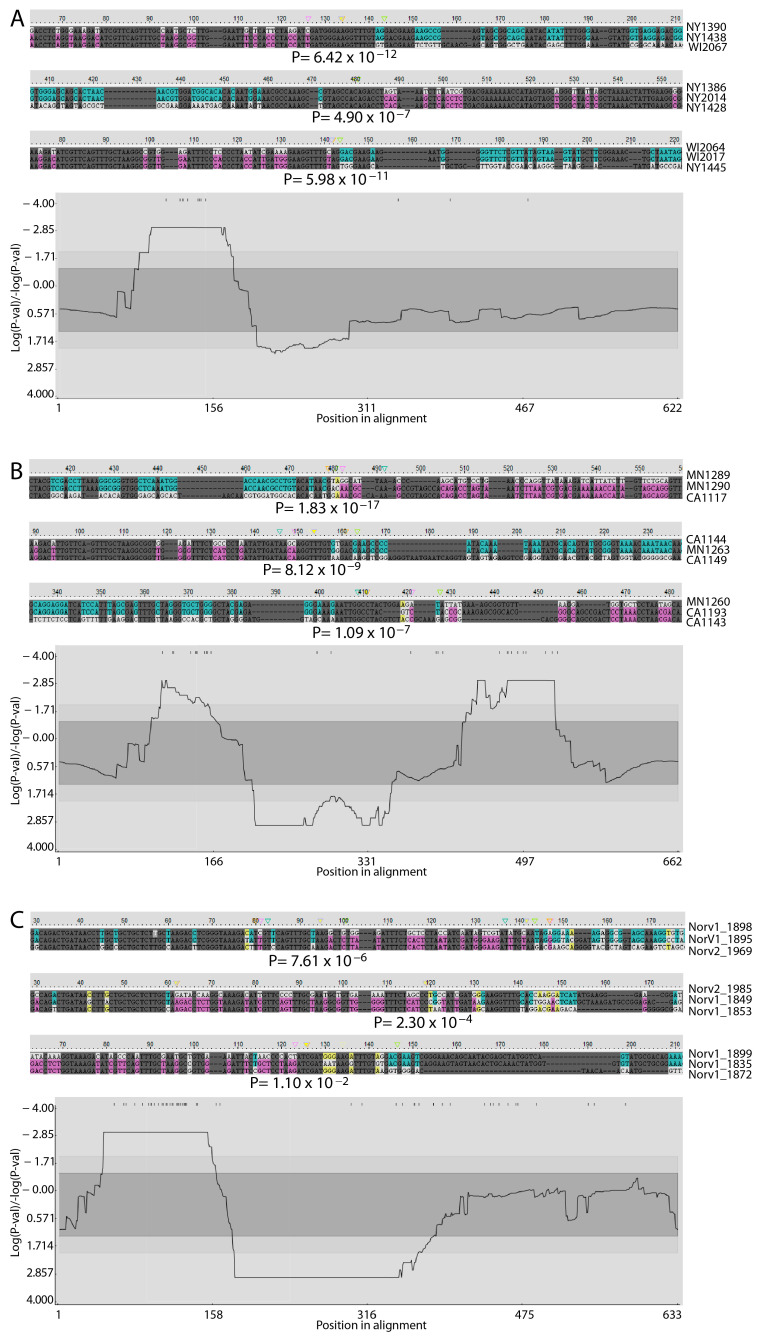
Predicted intra- and inter-strain recombination events between *msp2/p44* genes of *A. phagocytophilum* strains, based upon host. Evidence of recombination is provided for (**A**) human-derived strains, HZ2_NY and Webster_WI; (**B**) horse-derived strains, Horse1_CA and Horse1_MN; and (**C**) sheep-derived strains, NorShV1 and NorShV2. Three examples of areas of recombination between the genes of strains infecting the same host and predicted with a high probability are shown in each case. The top sequence of each alignment was predicted to be the “major parent” (contributing most of the sequence; major parent-specific sequences in blue), the bottom to be the “minor parent” (contributing less; minor parent-specific sequences in pink), and the center sequence is the predicted recombinant gene (sequences attributable to major or minor parents in the corresponding color). Nucleotides that match in the major and minor parents but differ from the recombinant are colored yellow. The positions of the recombination sites used in the analysis are shown above the plot, with the predicted breakpoint sites indicated by inverted triangles above each alignment (yellow triangles indicate sites predicted by GENECONV). Probability values are derived from the GENECONV analysis and indicated beneath the predicted breakpoint site. A breakpoint P-density plot is provided for each analysis, in which the plotted values for the alignment of all human, horse, or sheep isolate-derived sequences correspond to probabilities that recombination breakpoints are not significantly clustered. The central shaded areas indicate the 95% and 99% confidence intervals for the expected degrees of breakpoint clustering in the absence of recombination hot- and coldspots. The fasta sequences used in the alignments are provided in Appendix A. Statistical results for the full series of recombination and breakpoint analyses performed on all alignments for all combinations are presented in Appendix A.

**Figure 2 pathogens-14-00233-f002:**
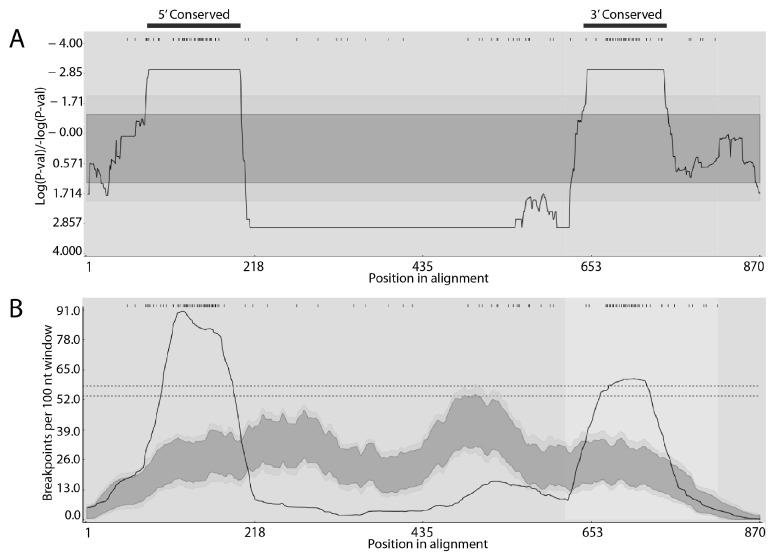
Predicted recombination events and their distribution among all *msp2/p44* genes of this study. (**A**) A breakpoint P-density plot of all predicted recombination events. (**B**) Predicted breakpoint site distribution. All 504 sequences are provided in Appendix A.

**Figure 3 pathogens-14-00233-f003:**
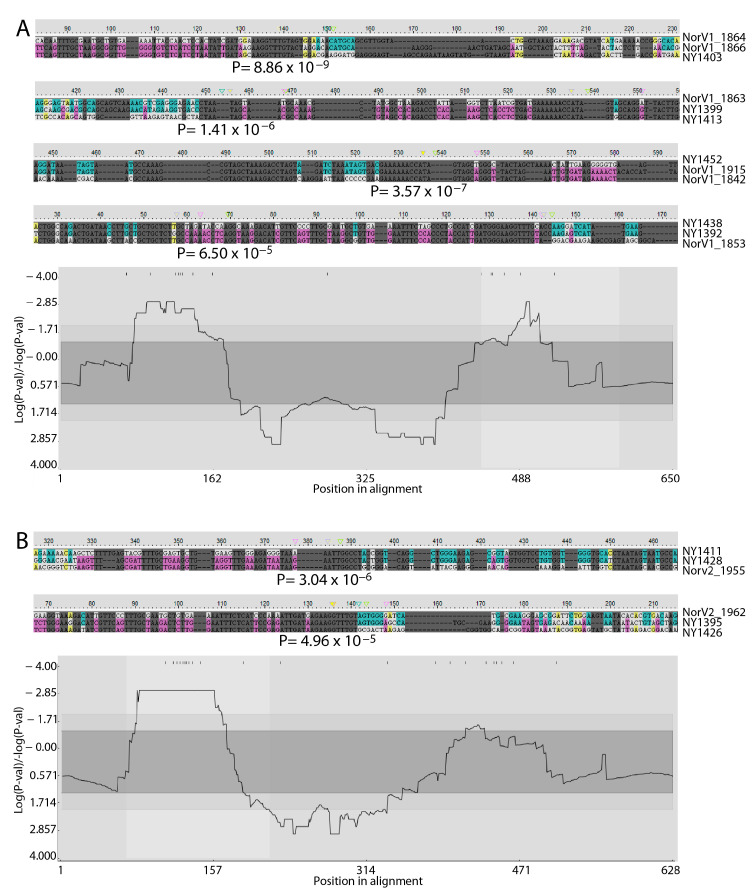
Predicted recombination events between individual HZ2_NY and NorShV1 or NorShV2 *msp2/p44* genes. Examples of recombination predicted with high probability between *msp2/p44* repertoires of (**A**) genes of the sheep strain-derived NorShV1 and human isolate-derived HZ2_NY and (**B**) genes of the sheep strain-derived NorShV2 and HZ2_NY. Breakpoint P-density plots are provided beneath each of the examples for the larger alignments from which each example was extracted. The methods used and the presentation of the results are as described for Figure 1.

**Figure 4 pathogens-14-00233-f004:**
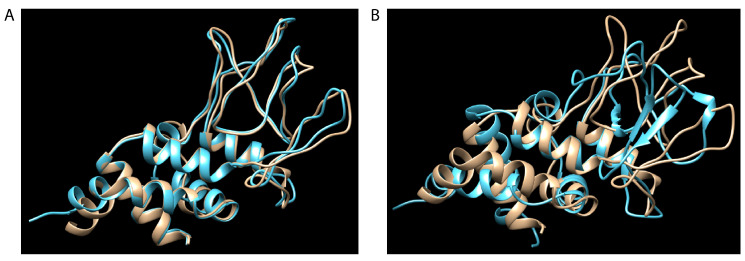
Superimposition of predicted structures of a recombinant MSP2 polypeptide and polypeptides encoded by its major and minor parent genes. (**A**) Structures of polypeptides encoded by genes HZ2_NY1445 (tan) and Webster_WI2017 (blue). (**B**) Structures of polypeptides encoded by genes HZ2_NY1445 (tan) and Webster_WI2064 (blue). In this example, HZ2_NY1445 is a predicted recombinant gene, Webster_WI2017 is the minor parent, and Webster_WI2064 is the major parent.

## Data Availability

All sequences employed in this project are provided in Appendix A and are found in GenBank accessions #CP006616, #LANS00000000, #FLMF00000000, #FLMC00000000, #CP046639, and #CP015376.

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
