# Peer review of "Generation of Population-Level Diversity in Anaplasma phagocytophilum msp2/p44 Gene Repertoires Through Recombination"

_pathogens, 2025, doi:10.3390/pathogens14030233_

Round 1
Reviewer 1 Report
Comments and Suggestions for Authors
Barbet et al. report on results of an analysis of genetic sequences in the msp2/p44 gene in the tick-borne pathogen, Anaplasma phagocytophilum. As they explain, msp2/p44 is an immunodominant surface protein that undergoes rapid antigenic variation during infections, which may help the pathogen escape sequestration by host immune responses. Antigenic variation is associated with shuffling of the many variant copies of the gene into an expression site within A. phagocytophilium' s genome. The investigators explored the question of whether there is evidence of recombination among the variant copies of the gene that would ultimately lead to more diversity among the many copies. Their results suggested that there are regions in the 5' and 3' flanking regions of the genes that appear to be areas of frequent (=hot spot) recombination, and that these recombination regions are shared among strains derived from the same hosts.
My comments relate to wanting a little more clarity around why the genomes analyzed? The authors report applying analyses (and subsequent results) to 2 human-derived genomes, HZ2_NY and Web_WI, 2 horse-derived genomes, Horse1_CA and Horse1_MN, and 2 Norwegian sheep-derived strains, NorShV1 and ApSheep_Norv2. However, in the Introduction, line 66, it is stated that there are 28 genome-sequenced strains of A. phagocytophilum. It isn't clear to me why the current analysis was limited to the 6 genomes used. Lines 122-133 in the Results section, appear to try to explain this in reference to results of a prior study (conducted by some of the same authors here) that examined the 28 genomes. In fact, these lines (122-133) in the Results section read like Materials and Methods and I question if they would be more appropriate there. I believe the authors are trying to say there was little genetic variation observed among the 28 genomes, which might explain why they focused on the 6 genomes of the current study. I think the study would benefit from a more explicit statement about that in the Materials and Methods section.
Additionally, I would request that the authors increase the font in the Figures. The axes are very difficult to read as is. Also, please include a key to the colors of the sequence changes in the legend of Figure 1.
Author Response
Barbet et al. report on results of an analysis of genetic sequences in the msp2/p44 gene in the tick-borne pathogen, Anaplasma phagocytophilum. As they explain, msp2/p44 is an immunodominant surface protein that undergoes rapid antigenic variation during infections, which may help the pathogen escape sequestration by host immune responses. Antigenic variation is associated with shuffling of the many variant copies of the gene into an expression site within A. phagocytophilium' s genome. The investigators explored the question of whether there is evidence of recombination among the variant copies of the gene that would ultimately lead to more diversity among the many copies. Their results suggested that there are regions in the 5' and 3' flanking regions of the genes that appear to be areas of frequent (=hot spot) recombination, and that these recombination regions are shared among strains derived from the same hosts.
[1] My comments relate to wanting a little more clarity around why the genomes analyzed? The authors report applying analyses (and subsequent results) to 2 human-derived genomes, HZ2_NY and Web_WI, 2 horse-derived genomes, Horse1_CA and Horse1_MN, and 2 Norwegian sheep-derived strains, NorShV1 and ApSheep_Norv2. However, in the Introduction, line 66, it is stated that there are 28 genome-sequenced strains of A. phagocytophilum. It isn't clear to me why the current analysis was limited to the 6 genomes used. Lines 122-133 in the Results section, appear to try to explain this in reference to results of a prior study (conducted by some of the same authors here) that examined the 28 genomes.
[2] In fact, these lines (122-133) in the Results section read like Materials and Methods and I question if they would be more appropriate there. I believe the authors are trying to say there was little genetic variation observed among the 28 genomes, which might explain why they focused on the 6 genomes of the current study. I think the study would benefit from a more explicit statement about that in the Materials and Methods section.
Author Response to points [1] and [2]: Concerns regarding choice of repertoires to analyze: We apologize for failing to make this point clearly in the original draft. We also agree with the reviewer about the location of this information within the manuscript. Accordingly, we have moved the information from the Results to the Methods section and have now devoted the initial section specifically to outlining the rationale driving the choices made (lines 93-116).
[3] Additionally, I would request that the authors increase the font in the Figures. The axes are very difficult to read as is. Also, please include a key to the colors of the sequence changes in the legend of Figure 1.
Author Response: We appreciate and agree with the reviewer’s concerns. The RDP5 software does not export images so all plots used in generating figures are screen-captures that included the native labeling. We have now manually created and superimposed X- and Y-axis labels, prepared in a much larger font size, to facilitate their viewing. This was done for Figures 1, 2, and 3. We have elaborated in the Figure 1 legend about the meaning of the nucleotide coloring seen in the recombination alignments (again a native output of RDP5), and have included additional explanation overall to facilitate reader understanding.
Reviewer 2 Report
Comments and Suggestions for Authors
Dear Authors,
The data obtained within the scope of the study are thought to contribute to the understanding of recombination among Anaplasma phagocytophilum isolates detected in different regions of the world. Considering that Anaplasma phagocytophilum is one of the most important tick-borne pathogens threatening human and animal health, the data obtained are considered to be very important.
Best regards
Abstract
In this section of the manuscript, the authors provide information about Anaplasma phagocytophilum and MSP2/P44 protein. In addition, brief information about the method used in the study and the results were given.
Introduction
The introduction of the manuscript was found to consist of a single paragraph. Given the importance of Anaplasma phagocytophilum, it is recommended to add more information about the pathogen (about its vectors and biology, clinical symptoms in hosts, strains, and their distribution) to the introduction.
Line 39. https://www.cdc.gov/anaplasmosis/stats/index.html address appears to be unavailable. It is suggested to review it again.
Line 40. Please add a comma before and.
Materials and methods
It was seen that detailed information was given about all of the methods applied within the scope of the study. The information provided in this section is written in a way that the reader can easily understand.
Results
It was observed that the data obtained within the scope of the study were given in great detail. This information was supported with figures.
Line 136 Please italicize A. phagocytophilum.
Discussion
In this section of the manuscript, the data obtained within the scope of the study are compared and discussed with the results of different studies. The information given in this section is considered to be sufficient.
Line 221 Please add a comma before and.
Line 271. Please do not italicize sp.
References
This section needs to be thoroughly reviewed and pathogen names and gene names should be italicized.
Author Response
[1] Dear Authors,
The data obtained within the scope of the study are thought to contribute to the understanding of recombination among Anaplasma phagocytophilum isolates detected in different regions of the world. Considering that Anaplasma phagocytophilum is one of the most important tick-borne pathogens threatening human and animal health, the data obtained are considered to be very important.
Best regards
Abstract
In this section of the manuscript, the authors provide information about Anaplasma phagocytophilum and MSP2/P44 protein. In addition, brief information about the method used in the study and the results were given.
Author Response: Thank you. No response needed.
[2] Introduction
The introduction of the manuscript was found to consist of a single paragraph. Given the importance of Anaplasma phagocytophilum, it is recommended to add more information about the pathogen (about its vectors and biology, clinical symptoms in hosts, strains, and their distribution) to the introduction.
Author Response: We thank the reviewer for providing this perspective. We have now expanded the Introduction section to include additional information regarding the pathogen’s biology, its transmission, clinical symptoms, strains, and their distribution. As most of these points are not central to the focus of the manuscript, however, we attempted to keep this brief while also fleshing out the background information somewhat. We hope this revised Introduction is now adequate.
[3] Line 39. https://www.cdc.gov/anaplasmosis/stats/index.html address appears to be unavailable. It is suggested to review it again.
Author Response: We thank the reviewer for pointing out this issue regarding accessibility of the CDC statistics site. We have updated the url for this site to match its new address.
[4] Line 40. Please add a comma before and.
Author Response: This comment is now moot due to our rewriting of the Introduction.
[5] Materials and methods
It was seen that detailed information was given about all of the methods applied within the scope of the study. The information provided in this section is written in a way that the reader can easily understand.
Results
It was observed that the data obtained within the scope of the study were given in great detail. This information was supported with figures.
Author Response: No response is needed.
[6] Line 136 Please italicize A. phagocytophilum.
Author Response: This correction has been made.
[7] Discussion
In this section of the manuscript, the data obtained within the scope of the study are compared and discussed with the results of different studies. The information given in this section is considered to be sufficient.
Author Response: No response is needed.
[8] Line 221 Please add a comma before and.
Line 271. Please do not italicize sp.
Author Response: These corrections have been made.
[9] References
This section needs to be thoroughly reviewed and pathogen names and gene names should be italicized.
Author Response: The references have all been converted into the MDPI format, reviewed, and pathogen and gene names are now all italicized.